# Traditional Conservation and Storage Methods for Ancient Chinese Painting and Calligraphy on Silk Manuscripts

**Wei Ren** [1,2,†] **and Na Cao** [3,*,†]

1   School of Artistic Design and Creation, Zhejiang University City College, Hangzhou 310015, China; wei.ren1012@foxmail.com
2   College of Architecture and Urban Planning (CAUP), Tongji University, Shanghai 200092, China
3   School of Art, Shandong Normal University, Jinan 250358, China
*   Correspondence: 620070@sdnu.edu.cn
†   These authors contribute equally.

**Abstract:** This study investigated traditional conservation and storage methods for Chinese silk manuscripts containing painting and calligraphy from the Warring States period (475–221 BC), the Qin dynasty (221–207 BC), the Han dynasty (202–8 BC; AD 25–220), and from the end of the Han to the present. At present, there is gap in the literature regarding the application of such methods to these works. The study methods include a literature review (classical and contemporary sources), expert interviews, and observation of traditional masters. The findings provide an improved understanding of the development of traditional technologies used for painting and calligraphy conservation since 475 BC. In this way, this work contributes to the body of knowledge regarding traditional conservation and storage methods, including mounting practices, scroll unfolding, and box storage.

**Keywords:** conservation; silk manuscript; painting; calligraphy; storage

## 1. Introduction

According to UNESCO (2020), "cultural heritage" includes traditions or living expressions inherited from one's ancestors and passed on to descendants. The knowledge and skills required to produce traditional crafts, for example, are a type of cultural heritage. Under this definition, the methods used to preserve and store ancient Chinese silk manuscripts containing painting and calligraphy can be considered a type of intangible cultural heritage.

Restoring (Chinese: 修复) and mounting (Chinese: 装裱) traditional calligraphy and painting require unique skills; in fact, it can take years to learn how to properly protect damaged paper or silk manuscripts (Chinese: 缣帛) (China Encyclopedia Press 1993). A silk manuscript is a type of manuscript that contains characters, images, and other symbols written or painted on silk fabric. Silk was an important material used for writing before the introduction of paper in China in approximately 100 BC (Clapperton 2014). The restoration and protection of silk-manuscript painting and calligraphy involve two basic principles—the first is to preserve the authenticity and integrity of the item; the second involves understanding how to protect the work and ensure its survival. The latter has been a longstanding problem in China with regard to mounting practices for works on silk.

In early China, characters were mainly written on bamboo slips (Chinese: 竹简), the preservation methods for which were relatively simple. Bamboo and wood were split into long, narrow pieces of varying length, and the surfaces were smoothly shaved. Bamboo slips were stored in cloth bags called *nang* (Chinese: 囊) or cloth wrappers called *zhi* (Chinese: 帙). Generally, *nang* was used to store paintings while *zhi* was used to store calligraphy. Silk manuscripts became available during the Warring States period. The preservation method for a silk manuscript is determined by its softness. Silk manuscripts

were folded and stored in a casket or suitcase (Chinese: 夶, 篋); over time, however, the silk would become creased. Moreover, since the silk pieces needed for a long piece of writing cannot be "tied" together into a volume, they easily become scattered. Given such problems, new and better ways of protecting and storing these manuscripts were gradually developed.

Based on the concept of intangible cultural heritage, it is important to study traditional methods for conserving and storing these ancient works of calligraphy and painting on silk. There has been extensive research on Chinese calligraphy and painting, including investigations of the traditional uses of scrolls (Chinese: 卷轴), albums (Chinese: 册页), and other mounting forms. Few studies, however, have specifically focused on traditional protection and storage methods. The present study aimed to address this research gap by examining preservation methods for painting and calligraphy relics from the Warring States period (475–221 BC), the Qin dynasty (221–207 BC), the Han dynasty (202–8 BC; AD 25–220), and from the end of the Han to the present. To this end, we reviewed the relevant literature (both classical and modern) and interviewed three known masters of Chinese silk-manuscript painting and calligraphy. This study's findings can enhance our understanding of the development of technologies used to preserve ancient Chinese painting and calligraphy on silk.

## 2. Literature Review

Recently, there has been increased research interest in conservation skills as an inherited intangible heritage, with a focus on traditional storage and preservation methods. With regard to research in the Chinese context, ancient Chinese literature can be challenging to understand for scholars who lack a basis in ancient Chinese knowledge. Further, traditional Chinese characters are quite different from simplified Chinese characters. Therefore, it can be difficult for international scholars to directly study the literature on traditional Chinese conservation methods. Nevertheless, some international scholars have attempted research in this area, such as Rogerio-Candelera (2014), who explored the relationships among science, technology, and cultural heritage. Meanwhile, a number of other studies have specifically focused on silk conservation. Agarwal Naveen and Farris (1997), for example, studied the effect of moisture absorption on the thermal properties of *Bombyx mori* silk fibroin films. Des Barker et al. (2007), meanwhile, investigated the conservation of artifacts and heritage materials, spanning both historical and scientific disciplines. Des Barker et al. (2007)' research provides an understanding of the deterioration mechanisms of artistic works. Yuan (2013) identified the degradation characteristics of the silk used during the Xi Han period. Similarly, Zhang et al. (2011) examined how heat and moisture promote the deterioration of raw silk.

Some researchers, such as Marwick (1982), have investigated restoration methods for historical paper documents. In the Chinese context, Catcher et al. (2017) examined the problems associated with paper reinforcement strip repairs on a set of four hanging calligraphic scrolls. Gao et al. (2016), meanwhile, applied a rheological model to Xuan paper, and results show that the Burgers model can demonstrate the creep behavior of Xuan paper perfectly. Under the certain humidity and temperature, viscoelastic strain, elastic strain, and plastic strain increased with the increase in stress in the range of 0.2–0.8 σmax, and the proportion of plastic strain and viscoelastic increased. However, the proportion of elastic strain decreased. Moreover, a number of researchers have specifically investigated the nature of Chinese manuscripts, such as Cohen (1998), who examined the research on the Dunhuang manuscripts. An increasing amount of research on the conservation of Chinese manuscripts was seen in the 1950s (van Gulik 1958; Needham and Wang 1956). Few recent international studies, however, have specifically examined the traditional conservation and storage methods used for silk-based Chinese painting and calligraphy work.

## 3. Methods

First, we reviewed both recent and classical literature related to traditional conservation and storage methods for Chinese silk manuscripts. Second, because the traditional skills investigated here are typically inherited from masters, the authors conducted interviews with three masters of traditional Chinese silk-manuscript painting and calligraphy between June and December 2019. These interviews provided the primary data for this study. Thirdly, the observation method was conducted in this research.

Today, there are very few people who have inherited these kinds of traditional skills; even specialized research institutions have few real experts in the field. Additionally, as mentioned above, it can be difficult for modern scholars to fully understand the original meanings of simplified Chinese characters. "Masters" in this area are generally older and reluctant to be interviewed; thus, we maintained the anonymity of the three interviewees. Each had more than 20 years of experience in the field; they either held doctorates and professorships or were practical experts in the field. The interview questions covered three main themes: (1) traditional conservation methods for Chinese painting and calligraphy on silk, (2) traditional storage methods for such work, and (3) the development and different characteristics of such methods up to the present. In addition to these three themes, other details related to the topic were discussed during the interviews. Notes were taken during the interviews.

The observation method can be helpful for learning and understanding traditional skills. It can also be used to verify descriptions of traditional skills found in the literature. Therefore, in addition to interviews, this study employed the observation method to see how masters used traditional skills, they are also the interviewee of this research. This was undertaken at the China Central Academy of Fine Arts, where conservation masters teach.

Lastly, we used a qualitative approach to analyze the findings. We categorized conventional conservation and storage skills into three groups according to different Chinese chronicles (Table 1) and then listed the notable dynasties.

**Table 1.** Chinese Chronicles.

| Ancient | Imperial | |
|---|---|---|
| Neolithic, c. 8500–c. 2070 BC | Qin, 221–207 BC | |
| Xia, c. 2070–c. 1600 BC | Han, 202 BC–AD 220 (Western Han, Xin, Eastern Han) | |
| Shang, c. 1600–c. 1046 BC | Three Kingdoms, 220–280 (Wei, Shu, Wu) | |
| Zhou, c. 1046–256 BC | Jin, 266–420 (Western Jin, Eastern Jin, Sixteen Kingdoms) | |
| Western Zhou | Northern and Southern dynasties, 420–589 | |
| Eastern Zhou | Sui, 581–618 | |
| Spring and Autumn | Tang, 618–907 (Wu Zhou, 690–705) | |
| Warring States | Five dynasties and Ten Kingdoms, 907–979 | Liao, 916–1125 |
| | Song, 960–1279 | |
| | Northern Song | Western Xia |
| | Southern Song | Western Liao |
| | Yuan, 1271–1368 | |
| | Ming, 1368–1644 | |
| | Qing, 1636–1912 | |

## 4. Findings

Based on the literature, archaeological findings, observation and interview results, the traditional conservation and storage methods for silk-based ancient Chinese painting and calligraphy can be divided into three groups according to dynasty.

### 4.1. Folding and Storage in Caskets and Suitcases during and before the Warring States Period

The earliest method for protecting silk manuscripts was a continuation of the *nang* and *zhi* used for bamboo slips. Here, silk manuscripts were simply folded and placed inside a lacquer casket for preservation. In 1942, the Chu Silk Manuscript (Chinese: 缯书) was discovered in the Zidanku tomb from the Warring States period (475–221 BC), east of Changsha, Hunan Province (An and Chen 1963). The Chu Silk Manuscript is the only pre-Imperial Chinese silk manuscript found to date. It contains text arranged diagrammatically, surrounded by pictorial illustrations (Li and Falkenhausen 2019). The manuscript was taken out of China and is currently held by a museum in the US. According to Zhimin and Chen Gongrou, when the Chu Silk Manuscript was discovered, it was packed in a bamboo suitcase (Chinese: 竹篋). It has been described as follows:

> The Chu Silk Manuscript is stored in a bamboo suitcase and folded properly . . . The bamboo suitcase has a cover, a half inch high, . . . eight inches long, four and a half inches in length. Both the cover and the bottom of the bamboo suitcase are made of bamboo silk. It is in the shape of a hexagon . . . and the inside is lined with thin silk. (An and Chen 1963)

> Moreover, the silk manuscript's "right and bottom [sides] are not sewn". (Xiong 1975)

This method of folding silk and storing it in a box is also found in the Han dynasty. Other tombs unearthed from the same period revealed the marginal protection of silk fabrics. In 1957, a stack of silk fabrics with a length of 30 cm, a width of 10–23 cm, and a thickness of 5–6 cm was unearthed in the Chu tomb excavated in Zuojiatang, Changsha (Xiong 1975). When discovered by archaeologists, the pile of silk fabric had been glued together with mud and was separated by a special fabric. One of the silk fabrics is a square brown brocade (Figure 1), with a remnant length of 19.9 cm and a width of 8.2 cm. Two brocades are sewn onto this piece of fabric. On one side of the brocade, there is a 0.8 cm length of yellow silk. Three characters, nǚ wǔ shì (Chinese: 女五氏), are sewn onto the silk fabric. A rectangular red seal is printed on the brocade, but the seal is incomplete as a result of damage to the fabric (Xiong 1975). This object demonstrates that during the Warring States period, people knew how to use other materials to protect the margins of silk fabrics by splicing and inlaying the edges. This method for protecting edges, which was an extension of sewing techniques used in daily life, was later applied in painting and calligraphy mounting practices.

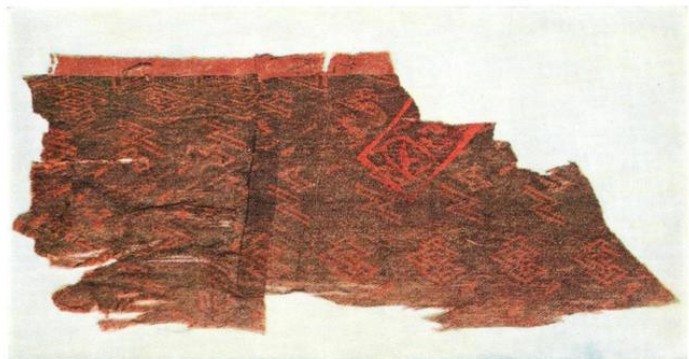

**Figure 1.** Remnant of a square brown brocade unearthed from the Zuojiatang Chu tomb in Changsha.

### 4.2. Painting and Calligraphy Scrolls during the Han Dynasty

Early methods for protecting silk-based calligraphy or painting were relatively simple and were not widely applied as a particular style. The appearance of a Han dynasty scroll marked the appearance of a formal mounting style. Specifically, a silk scroll with red lacquer was found in a Xi Han tomb in Changsha; however, it was broken after being unearthed, and

how it was folded is therefore unknown (Wang 1991). Nevertheless, this discovery showed that such scrolls appeared during the Xi Han period. Similarly, a silk manuscript from the Dong Han period had ink inscriptions describing the origin, length, weight, and price of silk. The inscriptions were written on both sides of the silk fabric, which was relatively long (Xiong 1975). In addition, maps drawn on silk were found in Tomb 3 of the Mawangdui site in Changsha (also from the Han dynasty), stacked in a rectangular lacquer casket (Mawangdui Group for Sorting Out Silk Books of Han Tombs 1975). The maps were unrolled and mounted to restore their original appearance. On one of the stitching bases, some of the silk pieces had dark marks on the hems, which were approximately 5 mm wide. The four sides of the map were folded. Because of the limited width of silk at that time, the map was made up of two left and right pieces (Mawangdui Group for Sorting Out Silk Books of Han Tombs 1975). These details reveal that during the Han dynasty, measures were taken to protect all four sides of silk paintings.

The "Yiwenzhi" section of the *Book of Han* (Ban 1990) explains that a scroll can be unfolded for reading and then rolled up and stored afterward for collection and preservation. The axis of the scroll is the center of rotation for painting or calligraphy on silk. Such a scroll is composed of at least two parts: the shaft rod (Chinese: 轴杆) and a silk carrier spread around it that contains the words or images. In addition to these examples mentioned by Wang (1991), other Han dynasty scrolls have been discovered at other archaeological sites.

Four silk paintings were found in Tomb 1 of the Mawangdui site. Each has a bamboo bar on the top for hanging. One is a T-shaped silk painting that covered the inner coffin (Figure 2). Two were hung on the east and west walls of the coffin chamber, and the other was stored in a rectangular lacquer casket that was unearthed from a box on the east wall (Hunan Provincial Museum 2004). The excavation report did not include information about how the two paintings were hung, but their shapes are similar to the T-shaped silk paintings (Figure 3) found in Tomb 3. These silk paintings are similar to those from the Warring States period, except that they are not depictions of daily life. Rather, they were used specifically for a funeral ceremony and had a particular theme and purpose.

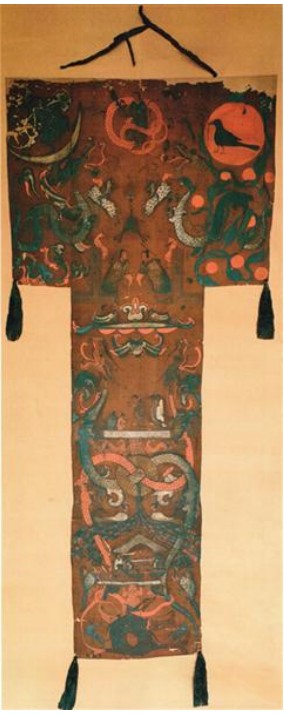

**Figure 2.** T-shaped silk painting from Tomb 1 in Mawangdui, Western Han dynasty.

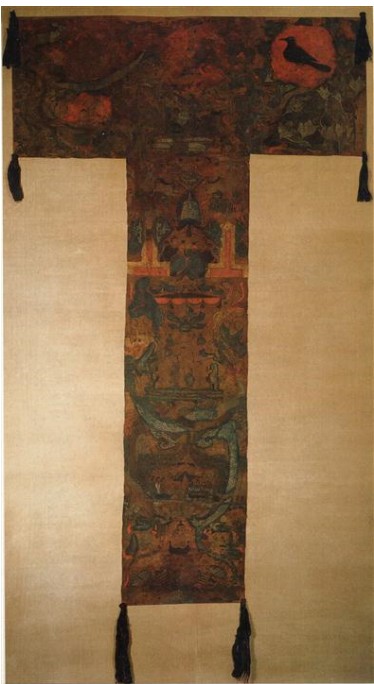

**Figure 3.** T-shaped silk painting from Tomb 3 in Mawangdui, Western Han dynasty.

Tomb 3 also yielded silk manuscripts found in another lacquer box on the east wall. We divided them into two types according to how they were preserved. The first manuscript is written on a full length of silk with a height of 48 cm. It was folded into a rectangle and placed in a grid under the lacquer box; the edge of the folding was broken. The second manuscript has a height of 24 cm. It had been rolled on a long strip of wood and was pressed under two rolls of bamboo slips, which had been fused together for a long time and were seriously damaged. (Figure 4) (Xiao 1974). From the preservation of the latter manuscript, we can observe the following: (1) The silk manuscript had been cut and selected specifically for its width. (2) When rolled on the long strip of wood, regardless of whether they were initially installed together, the method of rolling up the manuscript with the wood strip at the center was not a mistake and was obviously the owner's intention. (3) Putting the manuscript together with the two volumes of bamboo slips indicates that the depositor had owned it and saved it; this is fundamentally different in nature from the previous T-shaped silk paintings.

These new archaeological relics may overturn our current understanding of ancient scrolls in China. The existing literature, however, has confirmed early silk mountings from the Han dynasty. For example, the *Book of Han* manuscript, volume 30, shows that bamboo and silk were used for manuscripts during the Han dynasty. The terms *Jiedao* (Chinese: 介道), *Shou* (Chinese: 首), and *Mu* (Chinese: 目) identify the decoration of each part of the manuscript. Specifically, each volume of the manuscript had a different color, pointing to a type of scroll style widely used at the time. Thus, corresponding formatting standards and requirements for scroll conservation and storage were gradually formed (Fan 1973).

In the Ming-era book *Tongya*, Fang Yizhi summarizes the collection and distribution of manuscripts he reviewed in China from past dynasties. He counted "33,090 volumes in the Western Han dynasty and 13,269 volumes in the Eastern Han dynasty" (Fang 1988). Among these volumes, many were silk manuscripts. The *Book of Han* includes 43 volumes with 790 pictures of 53 families in the "Manuscript of War" section (53家兵书, 790篇文章, 图43卷), as well as 18 volumes with 92 pictures of 11 families in the "Shape of Right Soldiers" section (11家兵形势分析, 92篇文章, 图18卷) (Ban 1990). This reveals that, at that time, both manuscripts and pictures were preserved in the same way as scrolls.

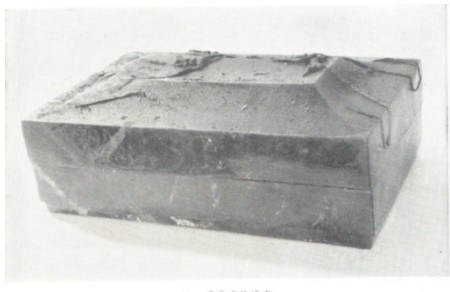

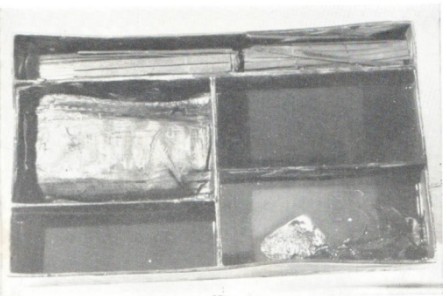

**Figure 4.** Lacquer casket found in Tomb 3 of Mawangdui from the Han dynasty (The English in the figure should be: The lacquer casket of silk manuscripts).

### 4.3. Painting and Calligraphy Scrolls after the Han Dynasty

While scrolls were used to mount silk manuscripts during the Han dynasty, we have not found specific evidence in the form of relics. Paintings on silk at that time served different purposes from the painting and calligraphy used in everyday life. After the Han dynasty, however, the appreciation and aesthetic function of calligraphy and painting changed. This change is confirmed by silk manuscripts and paintings that were unearthed later, as well as by the masters interviewed for this study.

The literature indicates that calligraphy and painting in the form of scrolls was widespread during the Wei and Jin dynasties (AD 220–420) and the Southern and Northern dynasties (AD 420–589). The Tang-era *Suishu Jingji Zhi* (Chinese: 隋书·经籍志) covers the collection and dispersion of manuscripts before the Sui dynasty (AD 581–618). Emperor Song Wu collected manuscripts as the spoils of war, totaling 4000 volumes. The characteristics of these volumes included a red axis, blue paper, and ancient, inelegant writing (Wei 1973).

The *Qimin Yaoshu* (Chinese: 齐民要术) by Jia Sixie, published during the late Wei dynasty, specifies precautions that should be taken when exhibiting or reading manuscripts. Specifically, a volume should be opened slowly; otherwise, it might break if it had not been folded carefully. Further, a bamboo belt should not be used to guide the manuscript open. If the manuscript is not dry, it will damage the first page and make a hole. The descriptions in this manuscript make it clear that the scroll style at the time included aspects such as the head of the scroll (Chinese: 天头), the first page (Chinese: 首纸), and a manuscript belt (Chinese: 书带).

The final establishment of the scroll form was essential for mounting silk-based calligraphy and painting. The scroll form was one of the three primary forms of mounting. This is evident in the appearance of various decorative elements, such as golden inscriptions and jade ornaments (Chinese: 金题玉躞) (Zhang 1963) and brocade embroidery (Chinese: 锦绣褾, 是指书画作品的包首) (Yang 1988). Decorative styles changed throughout the various dynasties; one example is the Xuanhe decoration hand roll (Chinese: 宣和装手卷). In this regard, Zhang Yanyuan of the Tang dynasty noted that each picture manuscript was unique, making it difficult to discuss specific techniques for handling them. Zhang also noted that when people see paintings in a manuscript, they will not take the time to finish reading the manuscript (Zhang 1963).

### 4.4. Less Common Forms of Manuscript Protection

There are additional, albeit less used, approaches to silk-manuscript conservation and storage. For example, the use of a screen (Chinese: 屏风) for mounting was relatively rare; few relics employing this technique have been unearthed. Screens were mainly made of silk and contained painting and calligraphy work; they were closely tied to the early development of painting and calligraphy in China. Screens were initially placed indoors to block the wind or for privacy. The "screen" here was a painting on silk fixed in a frame. The structure was used to stretch and level the silk, making it convenient for viewing as well as aesthetically pleasing. The frame and the paintings complemented each other. The screen-style calligraphy and painting works that still exist were transformed into scrolls by later generations. We do not know a great deal about the forms of screens used in earlier stages. However, it is generally believed that a screen displaying calligraphy and painting should be mounted on the back of the frame (Chinese: 裱褙). As Zhang (1989) noted,

> [I]t is said in history that the Jingjuan (Chinese: 经卷) and screen in the Qin and Han dynasties were all mounted on the back. They were probably mounted together with double-layer silk or covered with coarse hemp paper on the end to increase the thickness and fastness and facilitate display and circulation. This is perhaps the earliest screen mounting.

In the *Manuscript of Painting and Calligraphy Decoration*, Du Zixiong notes that these screens were used in Buddhist ceremonies in the Thousand-Buddha Cliff in Shandong Province. The ancient painting, Chuang fan (Chinese: 幢幡), would be mounted on the back of the screen, with a banner (Figure 5) on its side, in the shape of a hanging shaft (Du and Du 1993).

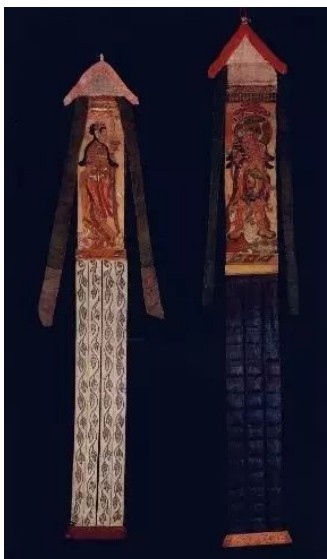

**Figure 5.** Chuang Fan.

Fengxia (2011) likewise notes that screens with calligraphy and painting were to be mounted on the back of the frame. Screens during the Tang were mostly made of coarse hemp paper, cloth, silk, and other rough materials for reinforcement (Fengxia 2011). However, screen painting had a significant disadvantage: without installation, it would have a short life (Fengxia 2011). There is insufficient evidence with regard to whether screens of calligraphy and painting were mounted and displayed during and before the Han dynasty. Rather, such silk-manuscript conservation and storage methods were used in later times.

## 5. Conclusions

This study investigated traditional conservation and storage methods for Chinese silk-manuscript painting and calligraphy work from the Warring States period (475–221 BC) to modern times. These traditional conservation and storage methods can be considered part of China's essential intangible heritage. Despite the limitations of ancient technologies, the archaeological evidence highlights the effectiveness of the adopted methods. Such methods have become a traditional skill that has been inherited and continued into the modern era.

This study addressed a gap in the existing research regarding traditional conservation and storage methods for ancient Chinese silk-manuscript painting and calligraphy. This study nevertheless has some limitations. Traditional skills are complex, and English-language scholarship has mostly been restricted to descriptions of technical details. Further, existing work on silk-manuscript conservation and on the Mawangdui site is relatively limited, and there are relatively few references in English. Thus, there was limited literature to draw upon.

This study contributes to the body of international knowledge on conservation and storage methods for silk manuscripts. There are still gaps, however, with regard to how such methods have been applied to paper manuscripts containing painting and calligraphy work. This can provide a direction for future research.

**Author Contributions:** N.C. and W.R. are the co-author for this paper, N.C. collected and analyzed the Chinese data. W.R. funded revised and final formulated the final paper. All authors have read and agreed to the published version of the manuscript.

**Funding:** This research was funded by Shanghai Planning Office of Philosophy and Social Science. Grant Number: 2018EGL002 and China Postdoctoral Science Foundation. Grant Number: 2019M661616.

**Institutional Review Board Statement:** Not applicable.

**Informed Consent Statement:** Not applicable.

**Conflicts of Interest:** The authors declare no conflict of interest.

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
