# Peer review of "Traditional Conservation and Storage Methods for Ancient Chinese Painting and Calligraphy on Silk Manuscripts"

_arts_

Round 1
Reviewer 1 Report
I think I understand the premise of this: that the evidence of format for the preservation of painting and calligraphy on silk (flat folded, scroll, screen) is demonstrated by historical finds in tombs during certain periods. However the writing style and repetition makes this a difficult read. The introduction is muddled and the title is used nine times in the first hundred lines.
Introduction
Lines 25/26, dates for the invention of paper
Lines 56-61, is a repetition of lines 26-31
Line 61, Why are the conservation masters anonymous?
Literature Review
List of references mean what? Need to be linked. Two important references missed: Van Gulik RH., Chinese Pictorial Art and Needham J., Science and Civilisation in China.
Methodology
Lines 101-104, Points 1 and 2 are the same
Very small sample group of 3 interviewees. How many were these chosen from?
Why only three?
Research Findings
Evolution from flat storage to rolled storage demonstrated by the addition of a silk margin. This much better and interesting but it must be remembered that removal from tombs tends to create lost context. Were these objects only used by the elite or by the general general population? These tombs are for whom and geographically where? A map might be useful to give the non Chinese reader insight as well as a time line for the dynasties.
Line 165, unrolled instead of uncoiled.
Line 170, demonstrated instead of proved.
Line 197, wood chips? Does that mean split wood to form a stave to the top?
Line 211, found (to be) different
Line 224, aesthetic
Line 243, cannot use this reference if it cannot be verified
Line 248, development (of the) history
Line 258, was entirely , no 'an'
Line 270, the wind or sight, no 'the'
Line 274, lacked physical materials means what?
Conclusion
Line 300, were they 'essential intangible skills', or something that developed over time? Evolution?
Line 303, inherited (and continued) in the modern era
Very minimal conclusion.
Author Response
I think I understand the premise of this: that the evidence of format for the preservation of painting and calligraphy on silk (flat folded, scroll, screen) is demonstrated by historical finds in tombs during certain periods. However the writing style and repetition makes this a difficult read. The introduction is muddled and the title is used nine times in the first hundred lines.
- The revised paper reduced the repetition terms of calligraphy and painting in the whole paper.
Introduction
Lines 25/26, dates for the invention of paper
- About dates for the invention of paper had been added, with reference. 100 BC in China.
Lines 56-61, is a repetition of lines 26-31
- I think you referred the repetition of lines 11-14. However, the lines 11-14 is the abstract, and lines 56-61 indicate the research aim.
- I revised Lines 56-61 too.
Line 61, Why are the conservation masters anonymous?
- We hope that the master can participate in this study anonymously, but one of the masters has some concerns, so he asked to be anonymous.
Literature Review
List of references mean what? Need to be linked. Two important references missed: Van Gulik RH., Chinese Pictorial Art and Needham J., Science and Civilisation in China.
- We added them, they are wonderful works on 1950s
Methodology
Lines 101-104, Points 1 and 2 are the same
- The Conservation Methods (point 1) and Storage Methods (point 2) are not the same.
- Conservation Methods focus on preservation, the minor physical restore (sometimes)
- Storage Methods focus on keeping the Silk Manuscript Painting and Calligraphy Work physical well.
Very small sample group of 3 interviewees. How many were these chosen from?
Why only three?
- We have added more details in this article about the reasons for how to choose, and why only three.
- At present, there are very few inheritors of this kind of traditional technologies, which is one of the important intangible cultural heritage in China. The inheritors of these heritages are generally older, So some inheritors are not particularly willing to be interviewed by us. Even in the national specialized research institutions, there are few real experts in this field.
- This research choose anonymous interviewees based on their work experience. These interviewers often work for more than 20 years,some of them have doctorates and professorships, or they are practical experts in this field.
Research Findings
Evolution from flat storage to rolled storage demonstrated by the addition of a silk margin. This much better and interesting but it must be remembered that removal from tombs tends to create lost context. Were these objects only used by the elite or by the general population?
- Only used by the elite. Only very few professionals able to access to such important cultural relics. The general public has no access at all. However, these experts may not be proficient in English, so there is few English literature research in this field
These tombs are for whom and geographically where? A map might be useful to give the non Chinese reader insight as well as a time line for the dynasties.
- We have added more detail owners and geographic information of these tombs in this paper. But it's very difficult for me to make a map.
Line 165, unrolled instead of uncoiled.
- Thanks, I changed
Line 170, demonstrated instead of proved.
- Thanks, I changed
Line 197, wood chips? Does that mean split wood to form a stave to the top?
- Just means the wood chips
Line 211, found (to be) different
- Thanks, I changed
Line 224, aesthetic
- Thanks, I changed
Line 243, cannot use this reference if it cannot be verified
- This is a very important book in China. Not every ancient Chinese book has an exact finishing time.
- Change it as the Jia Ancient Chinese Book, date unknown
Line 248, development (of the) history
- Thanks, I changed
Line 258, was entirely , no 'an'
- Thanks, I changed
Line 270, the wind or sight, no 'the'
- Thanks, I changed
Line 274, lacked physical materials means what?
- Deleted lacked physical materials
Conclusion
Line 300, were they 'essential intangible skills', or something that developed over time? Evolution?
- More detailed explanations on this
Line 303, inherited (and continued) in the modern era
- Thanks, I changed
Very minimal conclusion.
- I added more details.

Reviewer 2 Report
This is an interesting and useful paper, however some revisions for clarity of language, presentation and argument are required.
The grammar and language are not succinct and at times the writing becomes unclear. To assist the reader it would be helpful to have diagrams of various kinds of object and techniques that are discussed. The lack of clarity in the writing, the use of English words in uncommon ways and a need for more structural clarity from the Introduction, through the body of the paper to the Conclusion mean that diagrams, tables and illustrations would help considerably.
The paper would benefit from being passed to an editor for help with the language and grammar.
The references good be expanded (to include for example a link to UNESCO's definition of 'intangible heritage', and other literature relating to the topic of which there are further examples in the conservation literature.
The attached file has specific comments.
The paper is very interesting but at the moment its excellent potential is limited by the issues in presentation, writing and the flow of the argument.

Author Response
This is an interesting and useful paper, however some revisions for clarity of language, presentation and argument are required.
The grammar and language are not succinct and at times the writing becomes unclear. To assist the reader it would be helpful to have diagrams of various kinds of object and techniques that are discussed. The lack of clarity in the writing, the use of English words in uncommon ways and a need for more structural clarity from the Introduction, through the body of the paper to the Conclusion mean that diagrams, tables and illustrations would help considerably.The paper would benefit from being passed to an editor for help with the language and grammar.
- I asked the editors to review it before first submitting, and shall ask them to review it again if the paper been accepted.
The references good be expanded (to include for example a link to UNESCO's definition of 'intangible heritage', and other literature relating to the topic of which there are further examples in the conservation literature.
- Yes, I did
The attached file has specific comments.
- All the points noted in the PDF had been changed.
The paper is very interesting but at the moment its excellent potential is limited by the issues in presentation, writing and the flow of the argument.
- i am not good as draw a diagram, so try my best to explain it more clearly.

Round 2
Reviewer 1 Report
This is better and more coherent but the English is still not clear.
6 introduces rather than introduced
18 term 'cultural heritage' (no 'of')
24 Knowledge (not learning).... is a skill for students that also requires years of experience.
28 invention of paper, to be found in China about 100BC (ref)
29 There are two.... rework sentence
36 re-examines
39 furthers
44 bamboo slip length is
45 stored in different...... What? something missing
47 be a book
48 In the Chinese language Nang is...
53 creases
59 skills
62 this study aims...
63 explanation of
65 literature, including observing..
72 characters which are....
88-89 re-work sentence 'Although ...1960's
93-95 This information is not necessary here. Maybe in the author's notes at the end
107 there are very few people to inherit this kind of traditional technology.
108 Masters
110 chose
111 these interviewees...
114 questions covered three main themes..
119 detailed concerns
122 were normally recorded
126-134 Muddled. Please re-write
156 texts arranged
195 have appeared
203 Do you have a photograph of this map?
219 that were 57 lacquer and trousseau or tomb 57?
227 were unearthed
238 depositor had..
240 found to be different.
263 After the Han Dynasty, in daily life, the appreciation and the aesthetic function........changed.
325-327 Hanging belts are better described as banners
338 introduces
340 research shows
342 of valuable ancient Chinese heritage.
344 still keep their traditions
345 addresses
348 technical limitations
349 English journals are restricted to the explanation of technical details
352 there are not many references in English
Now to the content: Your research findings are fascinating. The conclusion is muddled and needs re-writing as it is weak.
4.4 screens. These have only just been mentioned here. I'm not sure if this fits in with the whole premise. You have been talking about folding, rolling and storage in boxes for the main body of the paper. This seems like an add on.
Author Response
To Reviewer
This is better and more coherent but the English is still not clear.
- Thank you so much for the comments and detailed suggestions, this paper had been re-edited before. However, if the paper can be formally accepted, we shall ask an English editor to review it again.
6 introduces rather than introduced
- Revised
18 term 'cultural heritage' (no 'of')
- Revised
24 Knowledge (not learning).... is a skill for students that also requires years of experience.
- Revised
28 invention of paper, to be found in China about 100BC (ref)
- Revised
29 There are two.... rework sentence
- Revised
36 re-examines
- Revised
39 furthers
- Revised
44 bamboo slip length is
- Revised
45 stored in different...... What? something missing
- Revised
47 be a book
- Revised
48 In the Chinese language Nang is...
- Revised
53 creases
- Revised
59 skills
- Revised
62 this study aims...
- Revised
63 explanation of
- Revised
65 literature, including observing..
- Revised
72 characters which are....
- Revised
88-89 re-work sentence 'Although ...1960's
- Revised
93-95 This information is not necessary here. Maybe in the author's notes at the end
Moved to Author Contributions
107 there are very few people to inherit this kind of traditional technology.
- Revised
108 Masters
- Revised
110 chose
- Revised
111 these interviewees...
- Revised
114 questions covered three main themes..
- Revised
119 detailed concerns
- Revised
122 were normally recorded
- Revised
126-134 Muddled. Please re-write
- Revised
156 texts arranged
- Revised
195 have appeared
- Revised
203 Do you have a photograph of this map?
- I think I can found the photo online, but I don't have the copyright to use it.
219 that were 57 lacquer and trousseau or tomb 57?
- It is not tomb 57, I revise it as to in a rectangular lacquer trousseau.
227 were unearthed
- Revised
238 depositor had..
- Revised
240 found to be different.
- Revised
263 After the Han Dynasty, in daily life, the appreciation and the aesthetic function........changed.
- Revised
325-327 Hanging belts are better described as banners
- Revised
338 introduces
- Revised
340 research shows
- Revised
342 of valuable ancient Chinese heritage.
- Revised
344 still keep their traditions
- Revised
345 addresses
- Revised
348 technical limitations
- Revised
349 English journals are restricted to the explanation of technical details
- Revised
352 there are not many references in English
- Revised
Now to the content: Your research findings are fascinating. The conclusion is muddled and needs re-writing as it is weak.
- Revised some sentence
4.4 screens. These have only just been mentioned here. I'm not sure if this fits in with the whole premise. You have been talking about folding, rolling and storage in boxes for the main body of the paper. This seems like an add on.
I think this would be fine to fit there due to the screens is also a form for Traditional Conservation and Storage Methods. If this paper does not mention this section, it would be quite uncomprehensive for this research.
